# Melatonin Enhances Drought Tolerance in Rice Seedlings by Modulating Antioxidant Systems, Osmoregulation, and Corresponding Gene Expression

**DOI:** 10.3390/ijms232012075

**Published:** 2022-10-11

**Authors:** Chengke Luo, Weifang Min, Maryam Akhtar, Xuping Lu, Xiaorong Bai, Yinxia Zhang, Lei Tian, Peifu Li

**Affiliations:** 1School of Agriculture, Ningxia University, Yinchuan 750021, China; 2College of Life Sciences, Northwest Normal University, Lanzhou 730070, China

**Keywords:** rice, drought stress, exogenous melatonin, antioxidant system, gene expression

## Abstract

Rice is the third largest food crop in the world, especially in Asia. Its production in various regions is affected to different degrees by drought stress. Melatonin (MT), a novel growth regulator, plays an essential role in enhancing stress resistance in crops. Nevertheless, the underlying mechanism by which melatonin helps mitigate drought damage in rice remains unclear. Therefore, in the present study, rice seedlings pretreated with melatonin (200 μM) were stressed with drought (water potential of −0.5 MPa). These rice seedlings were subsequently examined for their phenotypes and physiological and molecular properties, including metabolite contents, enzyme activities, and the corresponding gene expression levels. The findings demonstrated that drought stress induced an increase in malondialdehyde (MDA) levels, lipoxygenase (LOX) activity, and reactive oxygen species (ROS, e.g., O_2_^−^ and H_2_O_2_) in rice seedlings. However, the melatonin application significantly reduced LOX activity and the MDA and ROS contents (O_2_^−^ production rate and H_2_O_2_ content), with a decrease of 29.35%, 47.23%, and (45.54% and 49.33%), respectively. It activated the expression of *ALM1*, *OsPOX1*, *OsCATC*, and *OsAPX2*, which increased the activity of antioxidant enzymes such as superoxide dismutase (SOD), peroxidase (POD), catalase (CAT), and ascorbate peroxidase (APX), respectively. Meanwhile, the melatonin pretreatment enhanced the proline, fructose, and sucrose content by inducing *OsP5CS*, *OsSUS7*, and *OsSPS1* gene expression levels. Moreover, the melatonin pretreatment considerably up-regulated the expression levels of the melatonin synthesis genes *TDC2* and *ASMT1* under drought stress by 7-fold and 5-fold, approximately. These improvements were reflected by an increase in the relative water content (RWC) and the root-shoot ratio in the drought-stressed rice seedlings that received a melatonin application. Consequently, melatonin considerably reduced the adverse effects of drought stress on rice seedlings and improved rice’s ability to tolerate drought by primarily boosting endogenous antioxidant enzymes and osmoregulation abilities.

## 1. Introduction

In recent years, the phenomenon of drought stress has become more prominent due to global warming. Drought limits the growth and development of crops, hence affecting agriculture’s sustainable development. It is estimated that crop yield losses typically range from 30% to 90% under drought conditions [1,2]. Drought affects all aspects of a plant’s lifecycle. For instance, drought stress accelerates the aging process of plant leaves, causes chlorophyll to degrade faster, and reduces photosynthesis. This process produces an excessive accumulation of reactive oxygen species (ROS) in plants, which in turn causes oxidative damage to cell membranes, essential enzymes, proteins, and nucleic acids [1,3]. Therefore, it is crucial to balance ROS generation and scavenging under drought stress for normal plant growth and development. In particular, to successfully slow down ROS damage and preserve cellular redox equilibrium during the physiological response to drought, plants have developed a variety of morphological, physiological-biochemical, and molecular mechanisms [4].

Plants primarily activate enzymatic and non-enzymatic antioxidant mechanisms to remove an excessive accumulation of ROS. Superoxide dismutase (SOD), peroxidase (POD), catalase (CAT), ascorbate peroxidase (APX), glutathione peroxidase (GPX), and glutathione reductase (GR) are some of the antioxidant enzymes found in antioxidant systems. However, ascorbic acid (AsA), glutathione (GSH), carotenoids (Car), mannitol, anthocyanin, and alkaloids are the principal non-enzymatic antioxidants [5]. Plants have an activated intrinsic defense mechanism that begins to function only at a particular level of abiotic stress, such as drought stress [6]. Some exogenous chemical substances have been used in plants to reduce the adverse consequences of drought stress and successfully increase plant drought tolerance and adaptability. These chemical substances include osmoprotectants (e.g., glycinebetaine), antioxidants (e.g., glutathione), and plant growth regulators (e.g., melatonin). They play vital roles in reducing the adverse effects of drought stress and enhancing plant drought tolerance and adaptability [7,8,9].

Melatonin (MT), a growth regulator, has demonstrated significant potential for improving plant drought resistance [10]. Since 1995, melatonin has gained recognition as an effective strategy to increase crop tolerance against biotic and abiotic stresses, including drought, salt, low temperatures, high temperatures, heavy metals, nutritional deficiencies, and diseases [6,11,12]. For instance, melatonin pretreatment improved the tolerance of rice (*Oryza sativa* L.) to salt stress, because it increased rice’s dry and fresh weight while minimizing cell membrane damage [12]. Exogenous melatonin also effectively promoted cucumber (*Cucumis sativus* L.) seed germination under salt and water stresses [13,14]. In wheat (*Triticum aestivum* L.), a melatonin foliar spray reduced the damage caused by drought stress through increasing the antioxidant capacity, photosynthetic rate, and maximum efficiency of photosystem II (Fv/Fm) [1].

Under drought stress, melatonin application enhanced CAT, POD, SOD, and APX activities in maize (*Zea mays* L.) [15], buckwheat (*Fagopyrum tataricum* L.) [16], sage (*Salvia japonica* Thunb.) [17], and rapeseed [18]. Correspondingly, melatonin pretreatment upregulated the expression levels of these enzyme genes in chicory (*Cichorium intybus* L.), which may be the underlying molecular mechanism by which melatonin regulates enzyme activity [19]. Plant cell homeostasis is also controlled by the ascorbic acid-glutathione (AsA-GSH) cycle, which is directly mediated by melatonin [20,21]. For instance, as per the already published literature, melatonin reduced drought-induced leaf senescence in ‘Hanfu’ apple trees (*Malus domestica* Borkh.) by regulating the AsA-GSH cycle [21]. Tomato (*Lycopersicon esculentum* Mill.) seedlings treated with melatonin had increased GR production under drought stress [20]. As one of the antioxidant enzymes, GR has been proven to be involved in regulating the AsA-GSH cycle synergistically. Melatonin can increase chloroplast GR activity to maintain AsA homeostasis in drought-stressed plants [17]. Additionally, during drought stress, melatonin pretreatment increased the content of osmoregulatory components such as proline, soluble sugar, and soluble protein in soybean (*Glycine max* L.) [22]. 

Rice is the most water-consuming crop in many countries and regions. It uses 2-3 times more water than any other cereal [23,24]. Due to its high sensitivity to drought stress, rice production will inevitably be adversely affected by water shortages [25]. Therefore, increasing rice’s ability to withstand drought stress is crucial. As noted previously, exogenous melatonin treatment can successfully improve plant drought stress tolerance, mainly by promoting antioxidant systems [15,16,17]. Despite these results, it is largely unclear whether melatonin has a common or different mechanism for regulating plant tolerance. Its specific regulatory mechanism for the response of rice to drought stress is still unknown. In order to reveal the physiological and molecular mechanisms of how melatonin controls the response of rice to drought stress, rice seedlings’ physiological and molecular features, such as its ROS metabolism, antioxidant system, osmoregulatory substance accumulation and the corresponding gene expression, and melatonin synthesis gene expression, were examined under four treatments (control (CK), melatonin (CM), drought (DC), and melatonin plus drought (DM)). This study offers valuable insights into clarifying how melatonin improves rice’s drought tolerance.

## 2. Results

### 2.1. Melatonin-Dependent Improvement in the Growth Indices of Rice Seedlings under Drought Stress

As shown in Figure 1, there was no significant difference in the performance of rice seedlings sprayed with exogenous melatonin in both the CM treatment and the control (CK) under normal conditions (Figure 1A, top), which was reflected in the fact that the RWC did not differ between CM and CK (Figure 1B). In contrast, the drought-stressed rice seedlings showed extensive regions of leaf yellowing and withering, and fewer rice leaves stayed green in the DC treatment (Figure 1A, bottom left). The RWC and the root-shoot ratio of the drought-stressed rice seedlings in the DC treatment decreased by 24.03% and 14.38%, respectively, compared to the regular growing rice seedlings in CK (Figure 1B,C). However, these two growth indices of drought-stressed rice seedlings pretreated with exogenous melatonin in the DM treatment were substantially higher than those of drought-stressed rice seedlings in the DC treatment (Figure 1B,C), with an increase of 23.54% and 36.88%, respectively, demonstrating the effectiveness of exogenous melatonin at alleviating damaged leaves and maintaining more green leaves on the rice seedlings under drought stress (Figure 1A, bottom right).

### 2.2. Melatonin-Dependent Improvement in the Lipid Peroxidation and ROS Content of Rice Seedling Leaves under Drought Stress

Spaying melatonin did not alter the LOX activity, O_2_^−^ production rate, or MDA content of the leaves of rice seedlings in the CM treatment under normal circumstances (Figure 2A,C,D), except for a reduction in the H_2_O_2_ content (with a decrease of 40.19%) compared to CK (Figure 2B). However, drought stress significantly increased the levels of LOX activity, ROS (O_2_^−^ and H_2_O_2_) accumulation, and the MDA of rice seedling leaves in the DC treatment compared to that of CK, with an increase of 3.09-fold, 2.26-fold, 1.30-fold, and 2.38-fold, respectively (Figure 2A–D), which caused damage to the seedlings. An exogenous melatonin application significantly reduced the LOX activity, O_2_*^−^* production rate, H_2_O_2_ content, and MDA content of drought-stressed leaves in the DM treatment compared to that of the DC treatment, with a decrease of 29.35%, 45.54%, 49.33%, and 47.23%, respectively (Figure 2A–D), indicating that exogenous melatonin could mitigate the damage brought on by drought stress.

### 2.3. Melatonin-Dependent Improvement in the Antioxidant Enzyme Activity of Rice Seedling Leaves under Drought Stress

As shown in Figure 3, melatonin pretreatment significantly increased the SOD, POD, and CAT activities of rice seedling leaves in the CM treatment compared to CK under normal conditions, with increases of 2.68-fold, 1.10-fold, and 3.81-fold, respectively (Figure 3A–C), except for no obvious change in APX activity (Figure 3D). Furthermore, the activities of four antioxidant enzymes (SOD, POD, CAT, and APX) of rice seedling leaves were significantly higher in the DC treatment under drought stress compared to those in the CK treatment, with increases of 3.27-fold, 1.09-fold, 6.05-fold, and 1.39-fold, respectively (Figure 3A–D). Notably, an exogenous melatonin application further improved the antioxidant enzyme activities of drought-stressed rice seedling leaves in the DM treatment compared to those of the DC treatment, with increases of 27.62%, 24.41%, 18.47%, and 58.18%, respectively (Figure 3A–D), which aided in reducing the harm caused by excessive ROS to rice seedlings under drought stress.

### 2.4. Melatonin-Dependent Improvement in the Content of AsA and DHA of Rice Seedling Leaves under Drought Stress

Compared to CK, the melatonin pretreatment considerably increased the AsA content by 30.47% (Figure 4A), but did not affect the DHA content or the AsA/DHA ratio in the CM treatment (Figure 4B,C). Drought stress increased the DHA content by 11.38-fold (Figure 4B) and decreased the AsA/DHA ratio by 92.59% (Figure 4C), but it did not affect the AsA content of rice seedling leaves in the DC treatment compared to CK (Figure 4A). Unexpectedly, the exogenous melatonin application did not increase the ascorbic acid content, as evidenced by a reduction in the DHA content and the lack of apparent changes in the ascorbic acid content and AsA/DHA ratio of the drought-stressed leaves in the DM treatment compared to that of the DC treatment (Figure 4A–C). This suggests that ascorbic acid did not play a significant role in melatonin’s ability to relieve rice’s drought stress.

### 2.5. Melatonin-Dependent Improvement in the Proline, Sucrose, and Fructose Contents of Rice Seedling Leaves under Drought Stress

Proline is a perfect osmotic regulator, and it may also be employed in vivo as an oxidant, a membrane and enzyme protector, and a free radical scavenger. In plants, sucrose and fructose are essential energy sources, signaling molecules, and osmolytes. Figure 5 demonstrates that under typical growth conditions, spraying melatonin did not significantly change the rice seedling leaves’ proline, sucrose, or fructose levels in the CM treatment compared to CK. Under drought stress, the proline, sucrose, and fructose levels of rice seedling leaves significantly increased by 5.37-fold, 4.11-fold, and 1.27-fold, respectively, in the DC treatment compared to the CK treatment. Additionally, compared to CK, these three chemical concentrations were respectively enhanced by 1.76-fold, 2.19-fold, and 0.62-fold in drought-stressed rice seedling leaves pretreated with melatonin in the DM treatment (Figure 5A–C).

### 2.6. Melatonin-Dependent Improvement in the Expression Levels of Genes Encoding Physiological and Biochemical Substances of Rice Seedling Leaves under Drought Stress

One of the LOX synthesis genes, *OsLOX1*, three osmotic adjustment-related genes, *OsP5CS*, *OsSUS7*, and *OsSPS1*, and five antioxidant system-related genes, *ALM1*, *OsPOX1*, *OsCATC*, *OsAPX2*, and *OsVTC1-1*, were chosen and checked in rice seedling leaves under various treatments. As shown in Figure 6, an exogenous melatonin pretreatment did not alter the gene expression levels of *OsLOX1*, *OsP5CS*, *OsSUS7*, or *OsSPS1* in the CM treatment compared to CK. In contrast, drought stress dramatically increased these four gene expression levels in the DC treatment compared to CK, with an increase of 7.21-fold, 4.11-fold, 0.31-fold, and 0.20-fold, respectively. In response to drought stress, the melatonin application increased the expression of *OsP5CS*, *OsSUS7*, and *OsSPS1* by 73.21%, 31.26%, and 20.34%, respectively, in the DM treatment compared to in DC. However, the *OsLOX1* transcriptional level was unaffected by the melatonin application, which contradicts the decrease in LOX activity in the DM treatment compared to DC (Figure 6), indicating that *OsLOX1* may not be an essential gene regulating LOX synthesis.

SOD, POD, CAT, and APX are synthesized by the genes *ALM1*, *OsPOX1*, *OsCATC*, and *OsAPX2*, respectively. GDP-D-mannose pyrophosphorylase (GMPase), which is involved in the synthesis of AsA in rice leaves, is encoded by the gene *OsVTC1-1*. As can be observed from Figure 7, in comparison to CK, exogenous melatonin (CM treatment) and drought stress (DC treatment) clearly increased the gene expression levels of *ALM1*, *OsPOX1*, and *OsCATC* by 3.23~7.02-fold, 0.91~1.78-fold, and 2.38~2.95-fold, respectively. Drought stress (DC treatment) or exogenous melatonin (CM treatment) also significantly increased the gene expression levels of the *OsAPX2* and *OsVTC1-1* genes by 8.33-fold and 2.98-fold, respectively. The transcriptional levels of these five genes considerably increased in the DM treatment compared to the DC treatment, with an increase of 0.75-fold, 1.38-fold, 2.38-fold, 0.93-fold, and 4.68-fold, respectively, indicating that exogenous melatonin efficiently controls the antioxidant system by increasing the related gene expression levels in rice under drought stress.

### 2.7. Melatonin-Dependent Improvement in the Expression Levels of Endogenous Melatonin Biosynthesis-Related Genes of Rice Seedling Leaves under Drought Stress

It is now understood that melatonin is a signaling molecule. To investigate the effect of exogenous melatonin on the expression of the endogenous melatonin synthetase genes of rice seedling leaves under drought stress, two crucial rate-limiting enzyme genes, *TDC2* and *ASMT1*, were analyzed under various conditions in the melatonin synthesis pathway. The findings demonstrated that, compared to CK, exogenous melatonin (CM treatment) and drought stress (DC treatment) considerably increased the *TDC2* and *ASMT1* gene expression levels by 7-fold and 5-fold, approximately. Exogenous melatonin application significantly enhanced the transcriptional levels of these two genes in the DM treatment compared to the DC treatment, with an increase of 1.28-fold and 0.95-fold, particularly under drought stress (Figure 8), indicating that exogenous melatonin did indeed upregulate the expression levels of endogenous melatonin synthetase genes.

### 2.8. Correlation Analysis between Melatonin Biosynthesis-Related Genes and Antioxidant Enzyme Synthesis Genes

Exogenous melatonin also functions as an antioxidant to protect plants against intrinsic oxidative stress and drought. Gene expression levels were used to evaluate the link between the two to investigate the correlation between genes involved in melatonin biosynthesis and those synthesizing antioxidant enzymes under drought stress. The outcomes showed that the *R^2^* values between *TDC2* and the genes that make antioxidant enzymes (*ALM1*, *OsPOX1*, *OsCATC*, and *OsAPX2*) were, respectively, 0.919, 0.797, 0.876, and 0.753 (Figure 9A). Similar to this, there was a higher *R^2^* value between *ASMT1* and the genes that make antioxidant enzymes (*ALM1*, *OsPOX1*, *OsCATC*, and *OsAPX2*) (Figure 9B), indicating that every two genes between the genes related to melatonin biosynthesis and the genes that make antioxidant enzymes had a very significant positive correlation at the gene transcriptional levels under drought stress.

### 2.9. Construction of A Regulatory Model for Mitigating Effect of Exogenous Melatonin on Rice Seedlings under Drought Stress

We concluded that exogenous melatonin regulated the drought stress tolerance in rice according to the above research results. In-depth summaries of the intracellular physiological and biochemical alterations, changes in gene expression levels, and individual morphological changes associated with melatonin’s alleviation of drought stress in rice are provided (Figure 10).

The degradation of physiological and biochemical regulatory mechanisms in cells causes severe damage to the cell membrane under extreme drought stress, excessive ROS and NOS production accumulation, nucleic acid and protein degradation, increased MDA content, etc. This eventually results in cell death. Several defensive signaling cascades are often engaged when drought stresses plants, which further causes changes in gene expression and adjustments to the protein and energy metabolism in cells that have received stress signals. In this study, drought stress accelerated O_2_^−^ production, increased H_2_O_2_ and MDA content, and increased LOX activity, which caused rice seedlings to endure oxidative damage and membrane lipid peroxidation (Figure 10). In order to reduce oxidative stress and preserve the integrity of the cell membrane as much as possible, the stressed rice seedlings activated their antioxidant defense systems at this moment by upregulating the expression levels of antioxidant enzyme synthesis genes (*ALM1*, *OsPOX1*, *OsCATC*, and *OsAPX2*) and increasing the activity of antioxidant enzymes (SOD, POD, CAT, and APX) (Figure 10). Additionally, in stressed rice seedlings, osmolyte-like proline was accumulated by the upregulation of *OsP5CS* expression levels, and the content of energy metabolites such as sucrose and fructose was increased by the induction of *OsSUS7* and *OsSPS1* gene expression levels. These actions functioned to provide energy and maintain cell turgor pressure for plant growth (Figure 10).

*TDC2* and *ASMT1* gene expressions related to melatonin biosynthesis were significantly enhanced when exogenous melatonin was pretreated in rice seedling leaves under drought stress, resulting in increased levels of endogenous melatonin in rice cells. Endogenous melatonin increased the SOD, POD, CAT, and APX activities and the amounts of their respective genes’ in vivo expression (*ALM1*, *OsPOX1*, *OsCATC*, and *OsAPX2*) to either directly eliminate ROS by interacting with ROS, or indirectly scavenge ROS (Figure 10). The endogenous melatonin may then regulate the decline of membrane lipid peroxidation markers, such as LOX activity and MDA content, and further increase the proline, sucrose, and fructose accumulation by upregulating the expression levels of *OsP5CS*, *OsSUS7*, and *OsSPS1* genes in rice cells during drought stress (Figure 10). All of these might be the results of the physiological and biochemical alterations in vivo generated by the application of exogenous melatonin during drought stress and regulated by endogenous melatonin.

## 3. Discussion

Drought stress stunts plant growth by disrupting rice’s physiological metabolism and natural morphological structure [26]. As a versatile biomolecule, melatonin regulates physiological and biochemical processes, thereby improving the plant’s phenotype under drought stress [6]. A previous report showed that exogenous melatonin might boost soybean leaves’ RWC and increase their ability to withstand drought stress throughout their growth [27]. This study illustrated that rice seedlings displayed yellowing and wilting of leaves under drought stress, which was reflected by a reduced RWC and root-shoot ratio. Interestingly, exogenous melatonin relieved the drought stress in rice seedlings by turning the originally yellowing and wilting leaves back to green, and by increasing the leaves’ RWC and the root-shoot ratio. Similar findings were also verified in drought-stressed cotton (*Gossypium hirsutum* L.) [28] and potato (*Solanum tuberosum* L.) [29].

Drought stress disrupts the immune system’s natural ROS scavenging mechanism, which results in excessive ROS accumulation and lipid peroxidation, damaging cell membranes. Eventually, it impacts the growth and development of plants [30,31]. Our findings demonstrated that LOX activity, ROS (O_2_^−^ and H_2_O_2_), and MDA content considerably increased in rice seedling leaves under drought stress. Similar to our findings, Qi et al. [32] and Laxa et al. [33] found that the O_2_^−^ production rate and H_2_O_2_ content of plants increased during extreme drought stress, resulting in higher MDA content and an increase in membrane permeability. Several studies have shown that exogenous melatonin plays a vital role in reducing lipid peroxidation and reactive oxygen species in plants under abiotic stress [34,35]. For instance, after salt stress, applying exogenous melatonin could reduce ROS and MDA levels in cotton seedlings [34]. Qi et al. also found that exogenous melatonin reduced the level of membrane lipid peroxidation and protected the integrity of membrane lipids under high-temperature stress, inhibiting the MDA and ROS contents of chrysanthemum (*Chrysanthemum morifolium* ‘Jinba’), which improved the plant’s resistance to high-temperature stress [35]. This study also demonstrated a reduction in the LOX activity, ROS (O_2_^−^ and H_2_O_2_), and MDA levels after applying exogenous melatonin under drought stress, thus reducing membrane lipid peroxidation in rice seedling leaves. Both prior research and our study indicated that an exogenous melatonin application might successfully reduce the oxidative damage caused by drought stress, increasing the drought stress tolerance of rice.

Under typical circumstances, intracellular ROS are dynamically balanced and maintained at a lower level [31]. However, abiotic stress on plants disrupts the dynamic equilibrium of ROS in vivo, accumulating an excessive amount of ROS and cell membrane oxidative damage and ultimately resulting in oxidative stress [36]. At this time, plants start their antioxidant defense systems, including the production of enzymatic and non-enzymatic antioxidants, to eliminate excessive ROS [5]. Studies have shown that melatonin application certainly improves the ability of antioxidant defense systems to resist oxidative stress caused by abiotic stress [34,37]. For example, melatonin significantly increased the CAT, SOD, POD, and APX antioxidant enzyme activities in cotton roots, successfully slowing down the damage induced by salt stress and eventually improving salt tolerance in cotton seedlings [34]. Zhang et al. observed that exogenous melatonin promoted the accumulation of AsA and GSH in sugar beets (*Beta vulgaris* L.) under salt stress, in which the AsA-GSH cycle could operate swiftly and efficiently under the salt stress and sustain high antioxidant properties [37]. This study showed that exogenous melatonin dramatically upregulated the expression levels of the *ALM1*, *OsPOX*, *OsCATC*, and *OsAPX2* genes, improving the activities of the antioxidant enzymes CAT, SOD, POD, and APX, respectively. Consequently, exogenous melatonin relieved drought stress in rice seedlings by primarily strengthening enzymatic antioxidant systems rather than non-enzymatic antioxidants (AsA and DHA), which was somewhat different from our earlier research on the physiological mechanisms of exogenous melatonin alleviating alkaline stress tolerance in rice [38]. 

Proline and soluble sugars (such as sucrose and fructose) were accumulated in plant cells under drought stress and serve as osmolytes to maintain and protect plant macromolecules and structures from stress injury, eventually increasing the plant’s tolerance to drought stress [39,40]. A previous report showed that under salt stress, the application of 20 μM melatonin might accelerate the accumulation of osmotic compounds (such as proline and soluble sugars) in cotton seeds, improving the tolerance of seed germination to salt stress [41]. Our previous study indicated that exogenous melatonin increased the proline content while decreasing the sucrose and fructose levels in alkaline-stressed rice seedlings, which suggested that exogenous melatonin improved the alkaline tolerance by accumulating proline rather than sucrose and fructose in rice [38]. In the presented study, the application of exogenous melatonin increased the proline accumulation by activating *OsP5CS* gene expression and increasing the sucrose and fructose contents by upregulating the expression levels of the *OsSUS7* and *OsSPS1* genes in rice seedling leaves under drought stress. This finding suggests that melatonin may mediate the related metabolism of osmotic substances (proline, sucrose, and fructose), ultimately improving the drought tolerance of rice seedlings. 

Melatonin is regarded as a highly effective ROS scavenger and signaling molecule. Our findings first demonstrated that exogenous melatonin increased the expression levels of the melatonin synthesis genes *TDC2* and *ASMT1* (Figure 8). The increased expressions of these two genes likely contributed to the increased production of endogenous melatonin in rice. It is well known that melatonin scavenges too much ROS in two ways. One is that melatonin can interact with ROS directly to eliminate ROS. Garcia et al. estimated that one melatonin molecule might continuously interact with eight or more ROS molecules to maintain the optimal balance of ROS production to ROS scavenging [42]. In the current study, an exogenous melatonin pretreatment dramatically reduced the O_2_^−^ production rate and H_2_O_2_ content in rice seedling leaves under drought stress, which was likely because the absorbed and newly synthesized melatonin directly interacted with ROS to remove a fraction of the ROS, thus decreasing the O_2_^−^ and H_2_O_2_ contents in rice. 

As per an alternative way of ROS scavenging by melatonin, melatonin effectively reduces the accumulation of ROS by regulating the activity of antioxidant enzymes and the corresponding gene expression levels [43]. In this study, the application of exogenous melatonin enhanced SOD, POD, and CAT activities via regulating the expression levels of *ALM1*, *OsPOX1*, and *OsCATC*. The elevated antioxidant enzyme activities then scavenged ROS in rice seedling leaves under drought stress conditions, demonstrating that melatonin had an indirect role in eliminating ROS by activating antioxidant enzymes [44]. Additionally, based on the gene expression levels between melatonin synthesis genes (*TDC2* and ASMT1) and antioxidant enzyme synthesis genes (*ALM1, OsPOX1, OsCATC,* and OsAPX2), a substantial positive correlation between melatonin and antioxidant enzymes was established (Figure 7 and Figure 8). These findings reflect convectively on the hypothesis that melatonin scavenged ROS directly or indirectly via activating antioxidant enzymes, which is consistent with prior observations of melatonin [19,38].

## 4. Materials and Methods

### 4.1. Plant Materials and Growth Conditions

Zhonghua 11 (*Oryza sativa* L. cv. ‘Zhonghua No. 11’) variety of rice seeds that were full and uniform in size were washed with distilled water 5-6 times, disinfected with 15% NaClO for 30 min, and then dried with a filter paper. After five days of germination, the plantlets from the sterilized rice seeds were transferred to 96-well PCR plates with the bottoms of the tubes cut off. Afterward, the plantlets were grown in a greenhouse with rice nutrient solution under the following growth conditions: 28 °C/25 °C (day/night) and a 14/10 h (day/night) photoperiod. The nutrition solution was replenished every three days. The seedlings underwent experimental procedures after 21 days of growth. 

### 4.2. Experimental Design of Melatonin and Drought Stress Treatments

Melatonin (200 μM) was estimated to be the ideal treatment concentration for the phenotype of drought stress reduction, referring to our previous research [38]. The water potential under drought stress was set at −0.5 MPa, calculated using Michel’s formulae, and the drought stress was simulated using PEG-6000. The equation is as follows:*Ψ_s_* = −(1.18 × 10^−2^) *C* − (1.18 × 10^−4^) *C^2^* + (2.67 × 10^−4^) *CT*+ (8.39 × 10^−7^) *C^2^T*

where *Ψ_s_* refers to the water potential of the solution (bar), 1 bar = 0.1 MPa, C refers to the concentration of the PEG-6000 solution (g∙kg^−1^), and T refers to the temperature (°C).

Rice seedlings of uniform size, barely matured around 21 days, were chosen for 4 treatments. As seen in Figure 11, the following treatments were used on seedlings: (1) CK: seedlings were sprayed with double distilled water (ddH_2_O) on foliage without receiving a drought stress treatment; (2) CM: seedlings were sprayed with melatonin on foliage without drought stress treatment; (3) DC: seedlings were sprayed with ddH_2_O on foliage for three days and then received a drought stress treatment; and (4) DM: seedlings were treated with drought stress after being sprayed with melatonin on foliage for three days. Forty-eight plants were used in each of the four replicates for each treatment. For 7 days, either a regular nutrient solution (CK and CM) or a nutrient solution with PEG-6000 (DC and DM) was used to grow all stressed seedlings. Leaf samples from these treatments were then collected and stored at −80 °C for later use.

### 4.3. Measurement of Morphological and Physiological Indices

#### 4.3.1. Determination of Relative Water Content and Root–Shoot Ratio

Three seedlings were chosen at random, and their leaves were clipped off. These leaves were weighed for their fresh weight (FW) and then immediately immersed in test tubes with deionized water. After 24 h, the wet leaves immediately lost their water content. At this point, the leaves’ swelling weight (TW) was calculated. The leaves were then weighed as dry weight after being dried to a consistent weight at 70 °C (DW). The following equation was used to calculate the relative water content:RWC: RWC (%) = [(FW-DW)/(TW-DW)] × 100% 

Fifteen seedlings from each treatment were harvested for their cut roots and shoots. They were weighed after being dried to a consistent weight at 70 °C. The root DW to shoot DW ratio was used to represent the root-shoot ratio.

#### 4.3.2. Determination of Lipid Peroxidation and Reactive Oxygen Species Contents

As previously described, the MDA content was assessed using the thiobarbituric acid technique [38]. The O_2_^−^ production rate was calculated using the p-aminobenzene sulfonic acid method. Briefly, 0.2 g of rice leaf samples were homogenized with 65 mmol L^−1^ of phosphate buffer (PBS, pH 7.8) and centrifuged at 5000× *g* for 15 min at 4 °C. Then, 0.5 mL 10 mM hydrochloride, and 65 mM phosphate buffer were mixed with the supernatant and placed for 1 h. Subsequently, the above mixture was added with 17 mM p-aminobenzene sulfonamide and 7 mM α-theanine and incubated for 20 min. The absorbance of the supernatant was detected at 530 nm. The H_2_O_2_ content was obtained using the potassium iodide method. Briefly, 0.2 g of homogenized sample was mixed with ammonia, 20% TiCl_4_, and 95% hydrochloric acid and centrifuged at 10,000× *g* for 10 min at 4 °C. After repeatedly washing with acetone, the precipitate was dissolved with 2 mM H_2_SO_4_. The supernatant′ absorbance was measured at 410 nm.

The LOX activity was determined according to a previous study [45]. Briefly, 0.1 g of a rice leaf sample was homogenized with 5 mM sodium phosphate buffer (pH 7.5), 10 mM EDTA, 0.1% Triton X-100, and 5 mM β-mercaptoethanol and centrifuged at 4 °C and 16,000× *g* for 10 min. Then, the supernatant was added to 50 mM sodium phosphate buffer (pH 7.5), including 0.6 mM substrate (α-linolenic acid, dissolved in 100% ethanol). The absorbance was determined at 234 nm.

#### 4.3.3. Activity Assays of Antioxidant Enzyme

The crude enzyme solution was extracted using the Niu et al. technique [46]. Referring to the method of Li et al. [47], the SOD activity was measured using nitrogen blue tetrazolium as follows: The supernatant was added to a 3 mL reaction solution including 1.3 μM riboflavin, 13 mM methionine, 63 μM NBT, 0.1 mM EDTA, and a 50 mM PBS (pH 7.8) to undergo a chemical reaction at 30 °C for 15 min. The resulting solution was used to measure the SOD activity at 560 nm. The enzyme extract was reacted with a mixture of 20 mM guaiacol, 40 mM H_2_O_2_, and 100 mM PBS (pH 6.0) for the POD activity using the guaiacol method at 460 nm. The CAT activity was determined using a UV colorimetric method: the enzyme extract was added to a reaction solution containing 30% H_2_O_2_ and 0.15 M PBS (pH 7.8) to generate a solution to estimate the CAT activity at 240 nm.

#### 4.3.4. Analysis of AsA and DHA Content

The reduced ascorbic acid (AsA) and dehydroascorbic acid (DHA) contents were determined using our previous approach [38]. For AsA, the homogenized samples from 0.5 g of fresh rice leaf were mixed with 6% trichloroacetic acid (TCA, w/v) and were centrifuged at 16,000× *g* for 10 min at 4 °C. The supernatant was added with 200 mM PBS (pH 7.4), 10% TCA, 42% phosphoric acid (H_3_PO_4_), 4% 2,2′-bipyridine, and 3% ferric chloride under a water bath of 42 °C for 60 min. The absorbance at 525 nm was recorded. The DHA content was obtained by subtracting the AsA content from the total ascorbic acid content. For the total ascorbic acid, the supernatant was added to 6 mM dithiothreitol (DTT) and incubated for 15 min at 42 °C, followed by adding 0.4% N-ethylmaleimide (NEM), and then the solution was placed at 25 °C for 2 min. After that, the above reaction was mixed with 200 mM PBS (pH 7.4), 10% TCA, 42% phosphoric acid, 4% 2,2′-bipyridine, and 3% ferric chloride and incubated for 60 min at 42 °C. The absorbance was measured at 525 nm.

#### 4.3.5. Measurement of Free Proline, Sucrose, and Fructose Contents

The acidic ninhydrin colorimetric technique was used to measure the free proline concentration [48]. The sucrose and fructose contents were extracted using the Rosa et al. approach. The cardini et al. method was used to quantify the sucrose [49], while the Rosa et al. method was used to measure the fructose [48].

#### 4.3.6. Gene Expression Analysis

Using the Plant RNA Kit (TaKaRa, Dalian, China) and following the manufacturer’s instructions, the total RNA was isolated from leaf samples. The qualified total RNA was digested with gDNA Eraser (TaKaRa, Dalian, China) to eliminate genomic DNA. The PrimeScriptTM RT reagent kit (TaKaRa, Dalian, China) was then used to create the first-strand complementary DNA (cDNA) from the treated total RNA. According to the manufacturer’s instructions, qRT-PCR was carried out with a qTOWER3G Real-Time PCR System (Analytik Jena AG, Jena, Germany) using TB Green^®^ Premix Ex Taq™ II reagent (TaKaRa, Dalian, China). The experiment was replicated three times. The rice *OsActin* gene (LOC_Os03g50885) was used as the internal control. Based on the preceding techniques, the relative expression levels of the target genes were calculated [50]. The details of each gene-specific primer are presented in Table 1 and were taken from our previous study [38].

### 4.4. Statistical Analysis

Data for various experiments were statistically tested using SPSS 25.0 (IBM, Armonk, NY, USA) with a one-way analysis of variance (ANOVA).

## 5. Conclusions

Drought decreased the growth indices of RWC and the root-shoot ratio. However, it increased the ROS (O_2_^−^ and H_2_O_2_), MDA content, LOX activity, antioxidant enzyme (SOD, POD, CAT, and APX) activities, and osmolyte (proline, sucrose, and fructose) contents in rice seedlings, resulting in oxidative and osmotic stress. Exogenous melatonin up-regulated the melatonin synthesis gene (*TDC2* and *ASMT1*) expression. It activated antioxidant enzyme synthesis and osmotic substance accumulation to scavenge excessive ROS. It helped maintain cell turgor pressure, alleviating drought-stress damage to rice seedlings and ultimately enhancing drought tolerance. Consistent with these improvements, under drought stress via melatonin application, the relative expression levels of *ALM1*, *OsPOX1*, *OsCATC*, *OsAPX2*, *OsVTC1-1*, *OsLOX1*, *OsP5CS*, *OsSUS7*, *OsSPS1*, *TDC2*, and *ASMT1* were all up-regulated. These results increase our understanding of the melatonin regulation mechanism in rice during drought resistance.

## Figures and Tables

**Figure 1 ijms-23-12075-f001:**
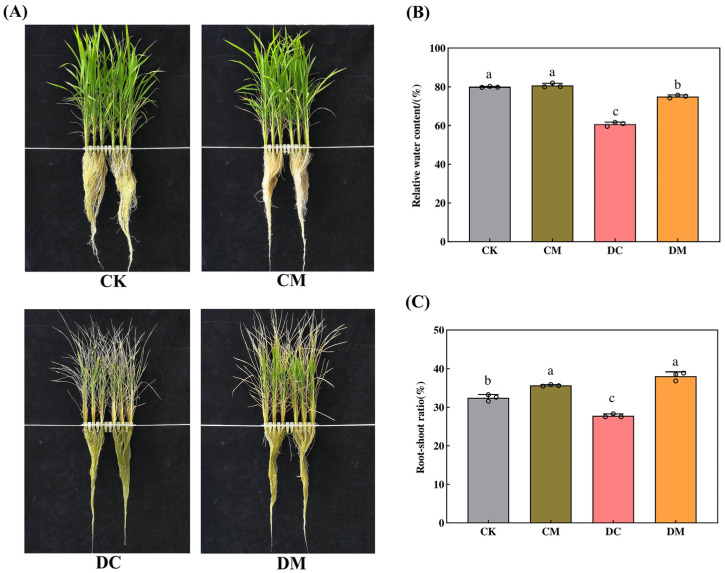
Effect of exogenous melatonin on rice seedling performance under drought stress. (**A**) Growth performance of rice seedlings under different treatments; (**B**) Relative water content; and (**C**) Root-shoot ratio. Data represent means ±SEs of three replicate samples. Different letters indicate significant differences (*n* = 3 and *p* < 0.05). CK, control; CM, control pretreated with melatonin; DC, drought stress; and DM, drought-stressed plants pretreated with melatonin. The same applies to the figures below.

**Figure 2 ijms-23-12075-f002:**
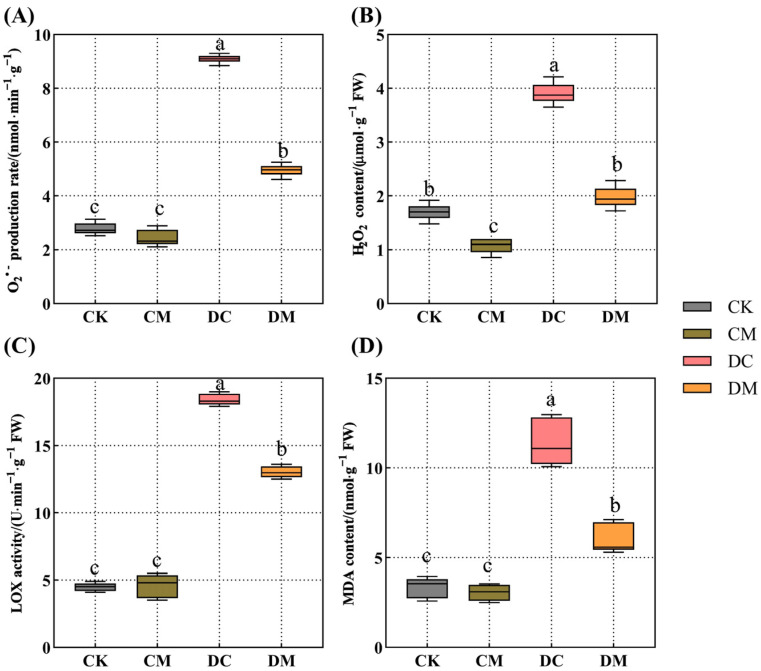
Effect of exogenous melatonin on lipid peroxidation and ROS under drought stress. (**A**) O_2_^−^ production rate; (**B**) H_2_O_2_ content; (**C**) LOX activity; and (**D**) MDA content.

**Figure 3 ijms-23-12075-f003:**
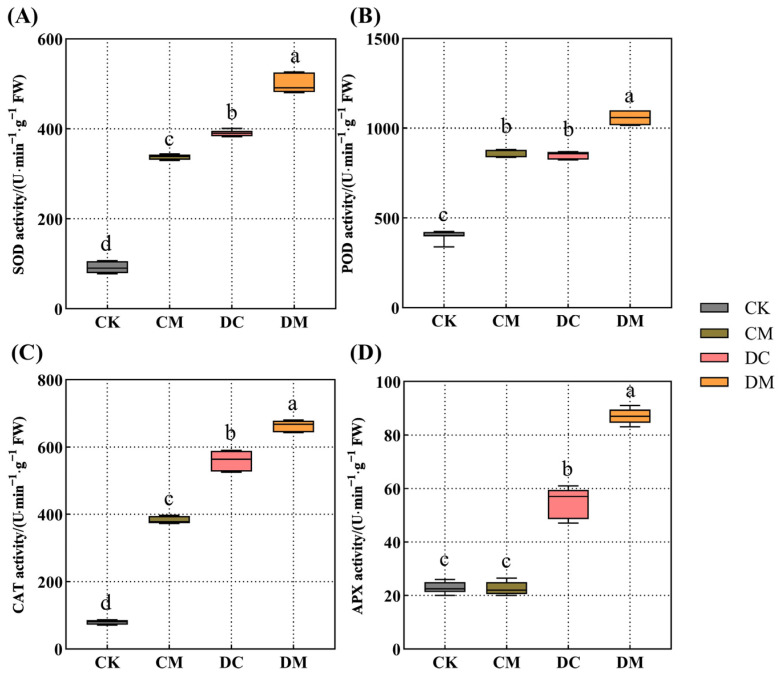
Effect of exogenous melatonin on antioxidant enzyme activity under drought stress. (**A**) SOD activity; (**B**) POD activity; (**C**) CAT activity; and (**D**) APX activity.

**Figure 4 ijms-23-12075-f004:**
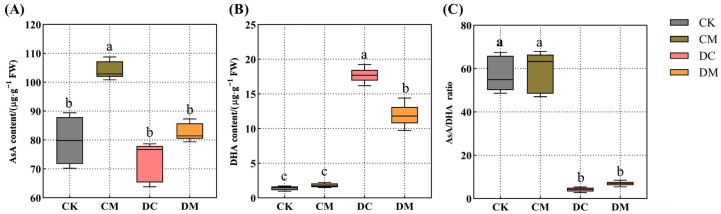
Effect of exogenous melatonin on ascorbic acid content under drought stress. (**A**) AsA content; (**B**) DHA content; and (**C**) AsA/DHA ratio.

**Figure 5 ijms-23-12075-f005:**
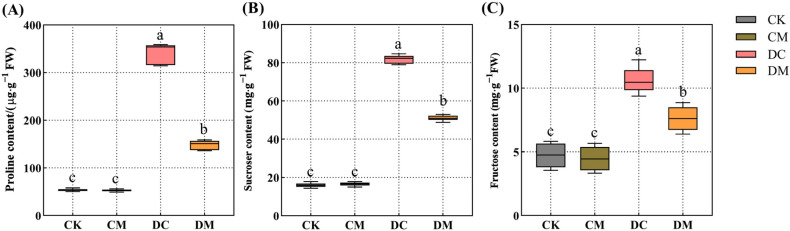
Effect of exogenous melatonin on proline, sucrose, and fructose contents under drought stress. (**A**) Proline content; (**B**) Sucrose content; and (**C**) Fructose content.

**Figure 6 ijms-23-12075-f006:**
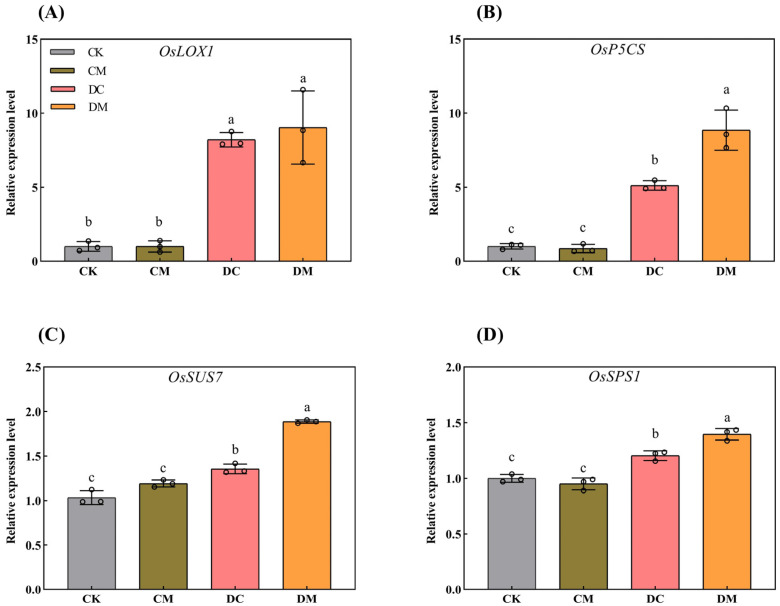
Effect of exogenous melatonin on LOX synthesis and osmotic adjustment-related gene expression levels under drought stress. (**A**) *OsLOX1* expression level; (**B**) *OsP5CS* expression level; (**C**) *OsSUS7* expression level; and (**D**) *OsSPS1* expression level.

**Figure 7 ijms-23-12075-f007:**
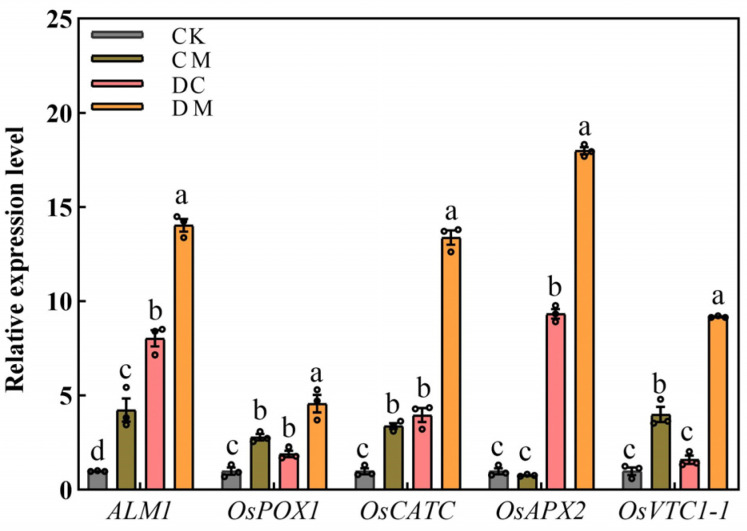
Effect of melatonin on antioxidant system-related gene expression levels under drought stress.

**Figure 8 ijms-23-12075-f008:**
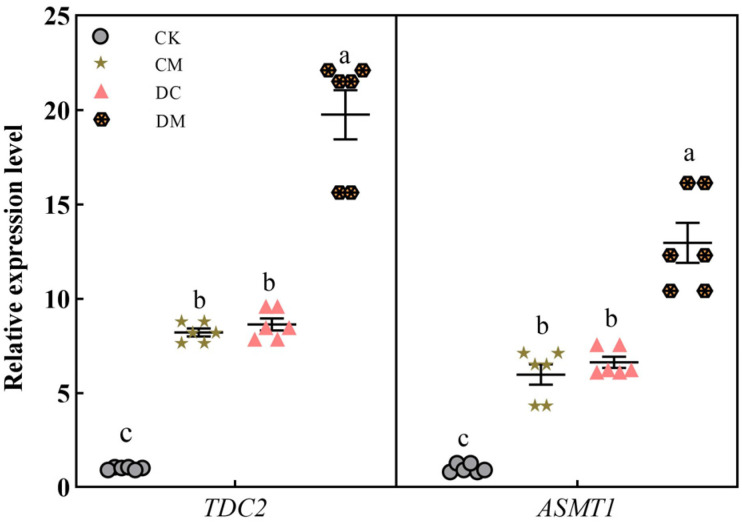
Effect of exogenous melatonin on melatonin synthetase gene expression levels under drought stress.

**Figure 9 ijms-23-12075-f009:**
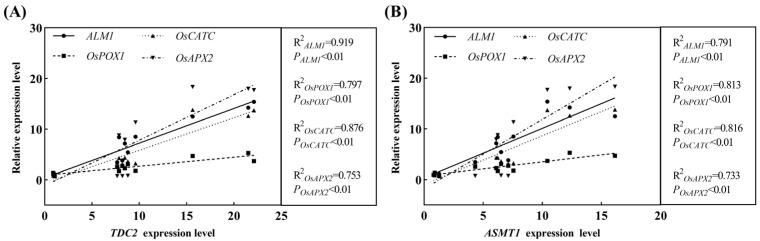
Correlation analysis of melatonin synthesis-related genes and antioxidant enzyme synthesis genes under drought stress. (**A**) Correlation of *TDC2* and antioxidant enzyme synthesis genes. (**B**) Correlation of *ASMT1* and antioxidant enzyme synthesis genes.

**Figure 10 ijms-23-12075-f010:**
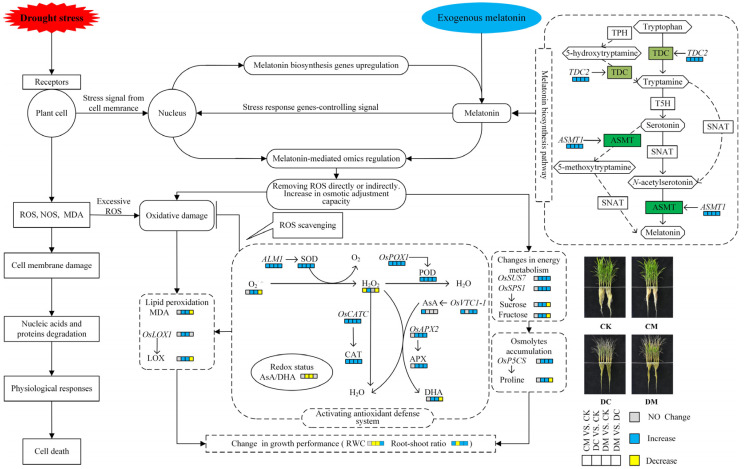
A proposed regulatory model for mitigating the effect of exogenous melatonin on rice seedlings under drought stress. The solid boxes represent the results of the well-documented studies, and the dashed boxes represent the results of this study. The diamond boxes represent substrates, intermediates, and final products in the melatonin synthesis pathway. The boxes represent synthetases, and the colored boxes represent two rate-limiting synthetases encoded by *TDC2* and *ASMT1* in this study.

**Figure 11 ijms-23-12075-f011:**
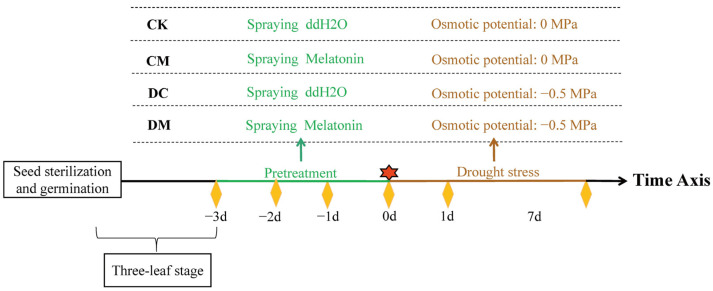
Diagram of spraying melatonin (refer to a previous report [12]).

**Table 1 ijms-23-12075-t001:** qRT-PCR primers used for gene expression analysis.

Gene Symbol	RGAP ID	Primer Sequence (5′-3′) (Forward/Reverse)	Product Size
*ALM1*	LOC_Os06g05110	CTGGCTGGGTTTGGCTTGT/TCGCCTGTCATCCTTGTAATC	158
*OsPOX1*	LOC_Os01g15830	TGCCTGTTGATGCTCTGCT/CCGCCTGTGCTACGATGGA	157
*OsCATC*	LOC_Os03g03910	ACAACCACTACGACGGCTTCA/CCTTGGCAATCACCACCTT	153
*OsAPX2*	LOC_Os07g49400	TTGTGAGTGGCGAGAAGGA/GGCGTAATCCGCAAAGAA	128
*OsVTC1-1*	LOC_Os01g62840	CAAGGGATTACATTACAGGC/TCAGGACCAATCAGACAGC	148
*OsLOX1*	LOC_Os03g49380	CTGACCCAAATACAGAAAGCA/GGGGAACACCCTCAACAATA	136
*OsP5CS*	LOC_Os05g38150	AATGACAGTTTAGCAGGAC/ACCACTATACAACCCATCC	87
*OsSUS7*	LOC_Os04g17650	TACAGGCACCAGATCCTAC/CTGCTGCTTGATTCTTTGA	200
*OsSPS1*	LOC_Os01g69030	GGCACAGCAAGACACTCCC/CGCCACGAACTAGACCATG	134
*TDC2*	LOC_Os07g25590	CAGAGTACCGACACCACCT/AACCCATAGCAAGGAACAA	104
*ASMT1*	LOC_Os09g17560	GCCAAGGCTCCCAGTAACAA/ACCTTTCCTCCAGCATCCC	179
*OsActin*	LOC_Os03g50885	GACCTTCAACACCCCTGCTA/ACAGTGTGGCTGACACCATC	114

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
