# Peer review of "Melatonin Enhances Drought Tolerance in Rice Seedlings by Modulating Antioxidant Systems, Osmoregulation, and Corresponding Gene Expression"

_ijms, 2022, doi:10.3390/ijms232012075_

Round 1

Reviewer 1 Report

Comments

In general, the submitted manuscript is set out to assess the ameliorative effect of Melatonin on rice seedling under drought stress. I trust that the paper has the publication potential, but it should be improved in various aspects, as mentioned in the following comments:

Title

Title should be modified as it does not contain the enough logic of the study.

Abstract

Line 13: The study not be consice to the one place. It should focus the global issue.

Line 15-16: „Nevertheless, the particular mechanism through which it helps alleviate rice is unknown.” this sentance should be re-written. and the next sentance should be started with „Therefore”.

Line 16-17: Please do mention that which kind of compound ormedium was used to employ the drought stress.

Line 17-18: „Subsequently, after observing their phenotypes, researchers looked into the metabolite compositions, enzyme activities”. This sentance should be modified, too.

Line 21: Please always explain the abbrevations at their first apperances e.g. ROS?

Line 21: Remove „was”

Line 22: Before MT, write down „However,”

Line 95: This study would offer……..

The sub-heading 2.1 should be replaced by „MT-dependent improvement in the growth indices of rice seedlings under drought stress”. This pattern should also be followed in the subsequent sub-headings but with different style. I trust that this way of presentation would attract more readers.

Line 123: Please do not mention the references in the results section.

There is a MAJOR flaw in presneting the results. The authors are advised to first explain the damage caused by drought stress and then its allevaition by MT application. Please always support your results by providing the qauntitative data e.g. drough has reduced the RWC by 40% and so on. You can explain them in fold changes as well e.g. MT improved the RWC by 2-fold. The way you have presented your results is very common and no longer useful since the reader can claerly see that from the figures. But, the new thing that you can offer to the reader is provide the quantitative data.

Line 231: It is not advised to ask a question. Please remove it with the appropriate statement.

Discussion

The information about the redox status is missing in this part. The authors have the data about ROS, ASA/DHA, APX, and CAT which are the contributors in the regulation of redox status of the plants under oxidative stresses. So, please provide 2-3 relevent snetances about redox alterations.

Line 437: Please provide the detailed method.

Conclusion is very poorly written even having so many attractive findings. The Please remove the first sentance of the conclusion. And first explain the effect of drought stress and then its mitigation by MT and the corresponding mechanistic pathway i.e. the inovment of genes, ROS and antioxidants.

Author Response

Point 1: Title should be modified as it does not contain the enough logic of the study.

Response 1: The title is "Melatonin enhances drought tolerance in rice seedlings by modulating antioxidant systems, osmoregulation and corresponding gene expression". See the red font in this paper for details..

Point 2: Line 13: The study not be concise to the one place. It should focus the global issue.

Response 2: Line 13 is modified as " Rice is the third largest food crop in the world, especially in Asia. Its production in various regions is affected to different degrees by drought stress". See the red font in this paper for details.

Point 3: Line 15-16: " Nevertheless, the particular mechanism through which it helps alleviate rice is unknown.” this sentence should be re-written. and the next sentence should be started with " Therefore”.

Response 3: Line 15-16 is modified as "Nevertheless, the underlying mechanism by which melatonin helps mitigate drought damage in rice remains unclear". The next sentence was started with " Therefore”. See the red font in this paper for details.

Point 4: Line 16-17: Please do mention that which kind of compound or medium was used to employ the drought stress.

Response 4: "Melatonin" was used to employ drought stress. Line 16-17 is modified as " Therefore, in the present study, the rice seedlings pretreated with melatonin (200 μM) were stressed with drought (water potential of -0.5mpa)". See the red font in this paper for details.

Point 5: Line 17-18: " Subsequently, after observing their phenotypes, researchers looked into the metabolite compositions, enzyme activities”. This sentence should be modified, too.

Response 5: Line 17-18 is modified as "These rice seedlings were subsequently examined for their phenotypes and physiological and molecular properties, including metabolite contents, enzyme activities, and corresponding gene expression levels". See the red font in this paper for details.

Point 6: Line 21: Please always explain the abbreviation at their first appearance e.g. ROS?

Response 6: The abbreviation at their first appearance were modified and explained. See the red font in this paper for details.

Point 7: Line 21: Remove „was”

Response 7: "Was" has been removed in Line 21. See the red font in this paper for details.

Point 8: Line 22: Before MT, write down „However,”

Response 8: "However" has been added in Line 22. See the red font in this paper for details.

Point 9: Line 95: This study would offer……..

Response 9: "It" has been replaced with "This study" in Line 95. See the red font in this paper for details.

Point 10: The sub-heading 2.1 should be replaced by „MT-dependent improvement in the growth indices of rice seedlings under drought stress”. This pattern should also be followed in the subsequent sub-headings but with different style. I trust that this way of presentation would attract more readers.

Response 10: The sub-heading 2.1~2.7 has been replaced by "MT-dependent improvement in…….. ” with different styles. See the red font in this paper for details.

Point 11: Line 123: Please do not mention the references in the results section.

Response 11: The References in the results section have been removed. See the red font in this paper for details.

Point 12: There is a MAJOR flaw in presenting the results. The authors are advised to first explain the damage caused by drought stress and then its alleviation by MT application. Please always support your results by providing the quantitative data e.g. Drought has reduced the RWC by 40% and so on. You can explain them in fold changes as well e.g. MT improved the RWC by 2-fold. The way you have presented your results is very common and no longer useful since the reader can clearly see that from the figures. But, the new thing that you can offer to the reader is provide the quantitative data.

Response 12: According to expert advice, we added quantitative data to support our results in this study. See the red font in this paper for details.

Point 13: Line 231: It is not advised to ask a question. Please remove it with the appropriate statement.

Response 13: The question sentence has been deleted and modified with the appropriate statement in Line 231. See the red font in this paper for details.

Point 14: The information about the redox status is missing in this part. The authors have the data about ROS, ASA/DHA, APX, and CAT which are the contributors in the regulation of redox status of the plants under oxidative stresses. So, please provide 2-3 relevant sentence about redox alterations.

Response 14: In fact, the information about the redox status was represented in discussion. The relevant sentence about redox alterations appear as follows: "Under typical circumstances, intracellular ROS are dynamically balanced and maintained at a lower level [40]. However, abiotic stress on plants disrupts the dynamic equilibrium of ROS in vivo, accumulating an excessive amount of ROS and cell membrane oxidative damage, ultimately, resulting in oxidative stress [41]. Plants start antioxidant defense systems including enzymatic and non-enzymatic antioxidants to eliminate excessive ROS and control their ability to withstand abiotic stress [42]". See the red font in this paper for details.

Point 15: Line 437: Please provide the detailed method.

Response 15: The detailed methods have been provided in Line 437. See the red font in this paper for details.

Point 16: Conclusion is very poorly written even having so many attractive findings. The Please remove the first sentence of the conclusion. And first explain the effect of drought stress and then its mitigation by MT and the corresponding mechanistic pathway i.e. the involvement of genes, ROS and antioxidants.

Response 16: The first sentence of the conclusion was deleted, conclusion was modified and perfected. See the red font in this paper for details.

Reviewer 2 Report

1. Luo and coauthors have investigated the role of melatonin in mitigating the adverse effects of drought stress in rice seedlings. The study seems to be a repetition of previous studies; as many studies are conducted to determine the role of melatonin against drought stress. https://doi.org/10.3389/fpls.2021.779382: This is an example of the previous studies and there are many more. I don't see any novelty in this work except rice crop. The manuscript is written well however, many changes are still needed.

2. Abstract section is written well, however, the rationale of the abstract section is poorly written. It must start with a problem and then the possible solution.

3. The Overuse of acronyms and abbreviations makes the manuscript difficult to read in many places. Therefore, I strongly recommend the authors to improve the readability of their text, starting from the abstract and all the way to the conclusions section.

4. The objective of the study (Subsequently, after observing their phenotypes, researchers looked into the metabolite compositions, enzyme activities, and corresponding gene expression levels in biochemical pathways) is poorly written and I strongly suggest the authors to improve it and make it attractive.

5. The conclusion at the end of the abstract section must also be improved and it should give a clear message to the reader.

6. The quality of the English language is poor just check the first sentence of the introduction: One of the major issues limiting plant growth, development and the ability to develop sustainable agriculture is drought, which is made worse by other factors such as global warming and the unsustainable use of water resources [1-2]. Therefore, I strongly suggest the authors to improve the quality of English language.

7. According to Cui et  al. [1] and Demidchik et al. [3] authors must avoid this type of statement in the manuscript as this is not a good way to write in my opinion.

8. The introduction must start with a problem statement why drought stress has increased in recent time. And what are the consequences of drought stress on plants ranging from germination, growth, physiological other processes and so on………………

9. The quality of the English language must be improved throughout the MS: likewise: What has to be focused on and researched by researchers is how to successfully increase plant drought tolerance and adaptability, as well as how to reduce the adverse consequences produced by drought stress.

This sentence is giving no clear message and authors should take care of this kind of things.

10. Osmoprotectants, exogenous antioxidants, and plant growth regulators are key to alleviating drought stress in plants: Same in case of this sentence no clear message to readers. Exogenous antioxidants??????

11. Line 62: a successful approach for it must be melatonin has been recognized as an important chemical to……………..

12. Stressors should be stresses

13. Line 67: (Oryza sativa L.) is given at not a proper place and it must be at the proper place.

Line 68-70: why authors are giving information on boron stress as a very large number of studies are available in the literature that has documented the role of melatonin in mitigating drought stress.

14. This section must include the role of melatonin to mitigate the adverse effects of drought stress. This must explain the mechanisms by which melatonin induced drought stress tolerance and what has left that must be explored.

15. Authors must add a clear hypothesis of their study and the objective of the study must also be cleared.

16. These lines: “two drought-related growth indicators, relative water content (RWC) and root-shoot ratio were chosen and studied in four treatments to ascertain the effects of MT on rice seedlings under drought stress” must be deleted from the text.

17. What was the need of this statement “similar to the findings of our prior study [29], drought stress greatly increased the levels of LOX activity, ROS (O2-, H2O2) accumulation, and MDA in rice seedling leaves (Figure 2A-D).

18. The results section is written in a very descriptive form therefore, I strongly recommend to authors to improve this section.

19. This statement “drought stress disrupts rice's physiological metabolism and natural morphological structure, which stunts the plant's growth” has again English language issue therefore, the quality of work starting from abstract right to the conclusion must be improved.

20. Line 291: Melatonin (MT) why melatonin here is written in this way??

21. There are some words that can never be suitable for scientific article writing. Likewise, the authors used exogenous MT administration and it must be an exogenous MT application.

22. The discussion section must be improved and it must be enriched with further logical reasoning.

23. Line 436-437: Analysis of ASA and DHA Content Using the Li et al approach, reduced ascorbic acid (ASA) and dehydroascorbic acid (DHA) levels were determined [59]. What is this? authors must explain the full protocol.

24. Authors must give full protocols for determination of lipid peroxidation and reactive oxygen species content, an antioxidant enzyme.

25. Finally from line 465 must be deleted.

26. The conclusion section is written poorly and it is giving no clear message. Therefore, authors must improve it and it should be based on study findings.

Author Response

Point 1: Luo and coauthors have investigated the role of melatonin in mitigating the adverse effects of drought stress in rice seedlings. The study seems to be a repetition of previous studies; as many studies are conducted to determine the role of melatonin against drought stress. https://doi.org/10.3389/fpls.2021.779382: This is an example of the previous studies and there are many more. I don't see any novelty in this work except rice crop. The manuscript is written well however, many changes are still needed.

Response 1: A growing number of studies have shown that melatonin is vital in alleviating plant drought tolerance. However, the mechanism of melatonin in alleviating drought damage to rice is unclear and needs to be revealed and researched. Therefore, it has important theoretical significance to rice production in global climate change.

Point 2: Abstract section is written well, however, the rationale of the abstract section is poorly written. It must start with a problem and then the possible solution.

Response 2: According to the advice of experts, we sorted out the causes and consequences of raising and solving the problem and revised and improved the abstract section. See the red font in this paper for details.

Point 3: The over use of acronyms and abbreviations makes the manuscript difficult to read in many places. Therefore, I strongly recommend the authors to improve the readability of their text, starting from the abstract and all the way to the conclusions section.

Response 3: According to the advice of experts, we replaced all acronyms with appropriate words from abstract to conclusion. See the red font in this paper for details.

Point 4: The objective of the study (Subsequently, after observing their phenotypes, researchers looked into the metabolite compositions, enzyme activities, and corresponding gene expression levels in biochemical pathways) is poorly written and I strongly suggest the authors to improve it and make it attractive.

Response 4: According to the advice of experts, we changed the sentence as follows: "Subsequently, theirs phenotypes, physiological and molecular properties such as metabolite contents, enzyme activities and corresponding gene expression levels were investigated". See the red font in this paper for details.

Point 5: The conclusion at the end of the abstract section must also be improved and it should give a clear message to the reader.

Response 5: The conclusion at the end of the abstract section has been modified as "Consequently, melatonin considerably reduced the adverse effects of drought stress on rice seedlings and improved rice's ability to drought tolerance by primarily boosting endogenous antioxidant enzymes and osmoregulation abilities". See the red font in this paper for details.

Point 6: The quality of the English language is poor just check the first sentence of the introduction: One of the major issues limiting plant growth, development and the ability to develop sustainable agriculture is drought, which is made worse by other factors such as global warming and the unsustainable use of water resources [1-2]. Therefore, I strongly suggest the authors to improve the quality of English language.

Response 6: According to the advice of experts, we modified the sentence as: "In recent years, the phenomenon of drought has become more prominent due to global warming. As is known to all, Drought stress limits the growth and development of plants, and then affect the sustainable development of agriculture [1,2]". See the red font in this paper for details.

Point 7: According to Cui et  al. [1] and Demidchik et al. [3] authors must avoid this type of statement in the manuscript as this is not a good way to write in my opinion.

Response 7: According to the advice of experts, we modified this type of statement in the manuscript. See the red font in this paper for details.

Point 8: The introduction must start with a problem statement why drought stress has increased in recent time. And what are the consequences of drought stress on plants ranging from germination, growth, physiological other processes and so on………………

Response 8: The introduction started with a problem statement. We also explained why droughts have become more severe in recent years. See the red font in this paper for details.

Point 9: The quality of the English language must be improved throughout the MS: likewise: “What has to be focused on and researched by researchers is how to successfully increase plant drought tolerance and adaptability, as well as how to reduce the adverse consequences produced by drought stress.” This sentence is giving no clear message and authors should take care of this kind of things.

Response 9: We modified the sentence as follows: " To reduce the adverse consequences produced by drought stress and increase successfully plant drought tolerance and adaptability, some exogenous chemical substance such as osmoprotectants (eg: Glycinebetaine), antioxidants (eg: Glutathione), and plant growth regulators (eg: Melatonin) were used to alleviate drought stress in plants and proved to play important roles [7-9]". See the red font in this paper for details.

Point 10: Osmoprotectants, exogenous antioxidants, and plant growth regulators are key to alleviating drought stress in plants: Same in case of this sentence no clear message to readers. Exogenous antioxidants??????

Response 10: We modified this sentence, which is the same as Response 9. Exogenous antioxidants refer to chemical substances such as Glutathione, Vitamin C, Vitamin E, and so on. See the red font in this paper for details.

Point 11: Line 62: a successful approach for it must be melatonin has been recognized as an important chemical to……………..

Response 11: According to the advice of experts, we modified this sentence in Line 62. See the red font in this paper for details.

Point 12: Stressors should be stresses

Response 12: "Stressors" has been replaced with "stresses" in Line 64. See the red font in this paper for details.

Point 13: Line 67: (Oryza sativa L.) is given at not a proper place and it must be at the proper place. Line 68-70: why authors are giving information on boron stress as a very large number of studies are available in the literature that has documented the role of melatonin in mitigating drought stress.

Response 13:  We've got (Oryza sativa L.) at the proper place. The reference [15] was misquoted and revised Line 68-70. See the red font in this paper for details.

Point 14: This section must include the role of melatonin to mitigate the adverse effects of drought stress. This must explain the mechanisms by which melatonin induced drought stress tolerance and what has left that must be explored.

Response 14: The mechanisms of melatonin induced drought stress tolerance were explained. Its underlying mechanism is assumed. See the red font in this paper for details.

Point 15: Authors must add a clear hypothesis of their study and the objective of the study must also be cleared.

Response 15: The hypothesis and objective of this study were added and cleared. See the red font in this paper for details.

Point 16: These lines: “two drought-related growth indicators, relative water content (RWC) and root-shoot ratio were chosen and studied in four treatments to ascertain the effects of MT on rice seedlings under drought stress” must be deleted from the text.

Response 16: According to experts' advice, we have deleted these lines from the text. See the red font in this paper for details.

Point 17: What was the need of this statement “similar to the findings of our prior study [29], drought stress greatly increased the levels of LOX activity, ROS (O2-, H2O2) accumulation, and MDA in rice seedling leaves (Figure 2A-D).

Response 17: We have deleted this sentence "similar to the findings of our prior study [29]". See the red font in this paper for details.

Point 18: The results section is written in a very descriptive form therefore, I strongly recommend to authors to improve this section.

Response 18: According to expert advice, we added quantitative data to support our results in this study. See the red font in this paper for details.

Point 19: This statement “drought stress disrupts rice's physiological metabolism and natural morphological structure, which stunts the plant's growth” has again English language issue therefore, the quality of work starting from abstract right to the conclusion must be improved.

Response 19: We modified this sentence as “Drought stress stunts the plant's growth by disrupting rice's physiological metabolism and natural morphological structure”. The quality of work, starting from the abstract to the conclusion, was modified. See the red font in this paper for details.

Point 20: Line 291: Melatonin (MT) why melatonin here is written in this way??

Response 20: We deleted “(MT)”and kept “Melatonin”. See the red font in this paper for details.

Point 21: There are some words that can never be suitable for scientific article writing. Likewise, the authors used exogenous MT administration and it must be an exogenous MT application.

Response 21: "Administration " has been replaced with "application". See the red font in this paper for details.

Point 22: The discussion section must be improved and it must be enriched with further logical reasoning.

Response 22: The discussion section was modified and enriched with further logical reasoning. See the red font in this paper for details.

Point 23: Line 436-437: Analysis of ASA and DHA Content Using the Li et al approach, reduced ascorbic acid (ASA) and dehydroascorbic acid (DHA) levels were determined [59]. What is this? authors must explain the full protocol.

Response 23: The complete protocols for measuring the reduced ascorbic acid (ASA) and dehydroascorbic acid (DHA) were provided and explained. See the red font in this paper for details.

Point 24: Authors must give full protocols for determination of lipid peroxidation and reactive oxygen species content, an antioxidant enzyme.

Response 24: The complete protocols for determining lipid peroxidation, reactive oxygen species content and an antioxidant enzyme were provided. See the red font in this paper for details.

Point 25: Finally from line 465 must be deleted.

Response 25: line 465 has been deleted.

Point 26: The conclusion section is written poorly and it is giving no clear message. Therefore, authors must improve it and it should be based on study findings.

Response 26: The conclusion section was modified and perfected. See the red font in this paper for details.

Round 2

Reviewer 2 Report

The authors have substantially improved the MS, therefore, it can be accepted for publication.

The quality of the English language is still poor, therefore, I strongly suggest the authors get it edited from English native speaker/English editing service by MDPI or any other.